# Preference of cesarean delivery and its associated factors among pregnant women attending ante natal care at public health facilities of Debrebrehan City, Ethiopia: Cross-sectional study

Lemlem Zewudu[1], Fetene Keshaun[2], Mulualem Silesh[2], Mitiku Tefera[2]*, Eyob Ketema Bogale[3], Aberham Demis[4], Zewedie Yeshaw Tekle[5]

1 Debre Berhan compressive specialized Hospital Amhara Region, Debre Berhan, Ethiopia, 2 Department of Midwifery School of Nursing and Midwifery, Asrat Woldeyes Health Science Campus, Debre Berhan University, Debre Berhan, Ethiopia, 3 Health Promotion and Behavioral Sciences Department, School of Public Health, College of Medicine and Health Sciences, Bahir Dar University, Bahir Dar, Ethiopia, 4 Department of Midwifery, Deber-Berehan Health Science College, Debre Berhan, Amhara, Ethiopia, 5 Department of Midwifery, Deber-Berehan Health Center, Debre Berhan, Amhara, Ethiopia

* mitikutefera889@gmail.com

## Abstract

### Background

A Caesarean section is a surgical procedure used to prevent or treat life-threatening maternal or fetal complications. Women's delivery preferences have become a global issue of interest to many researchers and clinicians, especially given the ever-increasing rate of cesarean sections. There is limited data on the preference for cesarean delivery and its associated factors for Ethiopian women, particularly in the study area. The aim of the study is to assess the preference for cesarean delivery and its associated factors among pregnant women attending antenatal care at public health facilities in Debre Berhan, Ethiopia, in 2023.

### Methods

An institution-based cross-sectional study design was done from May 5–20, 2023, among 512 participants, and a multi-stage sampling technique was used. The data were collected by using interviewer-administered semi-structured questionnaires. The data were entered by Epi Data version 4.6 and then transferred to SPSS version 25 for analysis. With logistic regression, those variables with a p-value <0.25 in the bivariate analysis were candidates for multivariate logistic regression, and variables with a p-value <0.05 were considered statistically significant.

**Data Availability Statement:** All relevant data are within the paper and its Supporting Information files.

**Funding:** The author(s) received no specific funding for this work.

**Competing interests:** The authors have declared that no competing interests exist.

## Result

The preference for a cesarean section was 26%, with a CI of 22.3% to 29.9%. Pregnant mothers who were not satisfied with their previous intrapartum care (AOR; 6.3 CI = (3.5–11), P = 0.01), had no knowledge about cesarean delivery (AOR; 2.9; 95% CI = 1.6–5.3), P = 0.01), had a previous history of spontaneous abortion (AOR; 3.1; 95% CI = (1.5–6.3), P = 0.001), lived in an urban area (AOR; 1.9; 95% CI = (1.0–3.5), P = 0.038), and had a current pregnancy-related problem (AOR; 4.8; 95% CI = 1.9–10), P = 0.001) were significantly associated with the preference for cesarean delivery.

## Conclusion

In this study, the preference for cesarean delivery was high as compared to the World Health Organization recommendation. Pregnant mothers who were not satisfied with their previous intrapartum care, had no knowledge about cesarean delivery, had a previous history of spontaneous abortion, had an urban residence, and had a current pregnancy-related problem were significantly associated with a preference for caesarean delivery. Clinicians who are working in the delivery room should improve their service provision by using patient-centered care to increase patient satisfaction.

## Introduction

A Caesarean section is a surgical procedure used to prevent or treat life-threatening maternal or fetal complications [1]. Pregnant women are normally involved in the decision-making process concerning the mode of delivery, and many factors affect their decision. These processes are influenced by a person's environment, values, personality, knowledge, and insight, which influence each other interactively [2].

Maternity delivery is one of the most important healthcare services in all countries [3]. The preference for cesarean delivery is defined as choosing a cesarean section as a mode of delivery [4]. Women's preferences for mode of delivery have emerged as a global subject of interest to many researchers and clinicians, especially with the steady increase in the rate of cesarean sections (CS), even though the World Health Organization (WHO) advises a maximum of 10–15% acceptable rate of cesarean sections [5]. Some data show cesarean section rates above 15% are not linked to further declines in maternal and neonatal mortality and morbidity [6].

The right to prefer a mode of delivery is a crucial component of compassionate and respectful care in modern obstetrics, as it fosters both maternal and neonatal well-being [7].

Cesarean delivery at the mother's request (CDMR) is a branch of elective cesarean sections performed not according to medical indication but at the mother's request [8].

C-sections have become a prominent indicator of progress in emergency obstetric care and a method to avert complications during labor and delivery [9]. Modern obstetric practice has seen increases in primary CS rates everywhere for medical, social, economic, and legal reasons [10]. For instance, studies have shown that Port Elizabeth has the highest CS rates (55.6%) [11], in Latin America (40.5%) [12] in the southern India region (32%) [13], in South Africa the CS rate is (42.4%) [14], and in Ethiopia also the CS rate is between 20.2 and 38.3% of mothers who have undergone cesarean section [15–17].

In Ethiopia, between 2000 and 2016, there was a slight increase in the national cesarean section rate from 0.7% in 2000 to 1.9% in 2016 [18]. Based on various attributes, differences

continued to exist. Compared to rural areas, which had a cesarean section rate of 0.9%, urban areas had a cesarean section rate of 10.6% in 2016 [18–20]. Determining their preferences in terms of mode of delivery will help reduce maternal and prenatal morbidity and mortality [19].

The advancement of delivery care, including cesarean delivery, has greatly improved the outcomes of birth globally, with a significant reduction in maternal morbidity and mortality. However, the evidence to support this is limited in Africa, especially Ethiopia, and more particularly in the study area. The study aimed to assess the factors influencing women's preferences for the cesarean section mode of delivery.

## Methods and materials

### Study design and settings

An institution-based cross-sectional study design was conducted in Debre Berhan town public health facilities from May 5 to May 20, 2023. Debre Berhan is located 130 km from Addis Ababa and 695 km from Bihar Dar, the capital city of Amhara Regional State. The total population of Debre Berhan town is 202,226; of the total population, 106,388 are females and 6,815 are pregnant mothers. According to the zonal health department report, Debre Berhan town has ten public health facilities, which are 2 government hospitals (of which are Debre Berhan comprehensive specialized hospital and Hakim Gizaw Hospital) and 8 health centers, which are Debre Berhan health center, Tebase health center, Ayer tena health center, Chacha health center, keyt health centers, Goshebado health center, Debre, and Enkulal Koso health center.

**Source of population.** All pregnant mothers who attended their ANC at a public health institution in Debre Berhan city.

**Study population.** All pregnant mothers who attended their ANC at selected health institutions in Debre Berhan City.

**Inclusion criteria.** All pregnant women who had one or more deliveries and who attended the ANC during the data collection period. In this study, the criteria for diagnosing a nonviable pregnancy for a first-trimester pregnant mother was urine HCG.

**Exclusion criteria.** Pregnant mothers who had previous CS scars.

**Sample size determination.** A single population proportion formula is used to calculate the sample size by considering the following statistical assumptions: P = proportion of the preference rate of CS among pregnant mothers from another study, 28.9%, Harar Regional State, Eastern Ethiopia [21].

(Z α/2 = Z score of 95% CI, d = Margin of error (5%). $n = \frac{\left(z_{\frac{\alpha}{2}}\right)^2 \times p(1-p)}{(d)^2}$.

n = (1.96)2 *0.289*0.711/ (0.05)2 = 316* 1.5 = 474 Then after adding 10% non-response rate, the sample size was 522.

### Sampling technique and procedure

A multi-stage sampling technique was used to select a representative sample. There are 10 public health facilities in Debre Berhan city; of these, 5 were selected using a simple random sampling method. The selected health facilities are Debere-Berhan Comprehensive Specialized Hospital (DBCSH), Debre Berhan Health Center, Tebase Health Center, Chacha Health Center, and Keyt Health Center. The sample was allocated proportionally for each health facility, and the allocation was done by using the average monthly ANC follow-up, which was 2100. Study participants were selected using a systematic random sampling technique. First,

determine the sampling interval (K) value by dividing the total number of pregnant women attending antenatal care during the study period by the total sample size, which gives $2.01 \approx 2$.

The probability allocation sampling technique was to select (nf x n)/N = (Sample final * number of total pregnant women attending antenatal care in each health facility/number of total pregnant women attending antenatal care within two weeks.

Where N is equal to 1050.

- Debre Berhan comprehensive specialized hospital = 522x 400/ 1050 = 198

- Debre Berhan Health Center = 522 x 220/ 1050 = 110

- Tebase Health Center = 522 x 150/1050 = 74

- Cacha Health Center = 522 x 176/ 1050 = 88

- Keyt Health Center = 522 x 164/1050 = 81 (Fig 1)

## Data collection methods

Data was collected using semi-structured questionnaires adapted from a review of relevant literature [22–26]. All questions were written in English and translated into Amharic (the local language) and then back to English by two different language experts to check for consistency and clarity. The questionnaire is divided into six sections (1–5) to obtain data on the socio-demographic characteristics of the respondents, obstetric factors, knowledge towards cesarean delivery, attitude towards cesarean delivery, and preference of cesarean delivery parts (see **S1 File**).

## Variables of the study

**Dependent variable.**   Preference of cesarean delivery.

**Independent variable.**   Socio demographic factors, obstetric factors, knowledge towards cesarean delivery, attitude towards ceesarean delivery and maternal satisfaction on previous intra partum care.

## Operational definitions

**Preference for cesarean delivery** implies the patient's choice of cesarean delivery without any fetal or maternal indication [27].

## Maternal knowledge

To assess maternal knowledge of cesarean delivery, the questions are adapted.1 point was given for each correct response, and 0 points for incorrect and 'I don't know' answers. The overall maternal knowledge score was described as good (7–10), intermediate (4–6), and poor (0–3) [26].

**Attitude towards cesarean delivery.**   The questionnaire for attitude assessment was served in a Likert scale format with strongly agree (score 5), agree (score 4), neutral (score 3), disagree (score 2), and strongly disagree (score 1).

Attitude toward CD was assessed with 10 statements for cesarean delivery. A median attitude score was computed for each respondent for all the statements to find the overall attitude of women towards that mode of delivery. A median attitude score of 3 or less was considered a negative attitude, and a score of more than 3 was considered a positive attitude toward that particular mode of delivery [28].

**Maternal satisfaction** is the satisfaction of mothers during service delivery. The level of satisfaction was assessed on a 5-point Likert scale (1, very dissatisfied; 2, dissatisfied; 3, neutral; 4,

# Debre Berhan town has 10 Health facilities

## Simple random lottery method used

| DBCSH | CHHC | DHC | KHC | THC |

## Proportional allocation used for each hospital (nf= n×Ni/N)

nf=198    nf=88    nf=110    Nf =81    nf=74

## Systematic RS

522

**Fig 1. Schematic representation of sampling procedure for determining the preference of C/S and its associated factors among pregnant mother attending ANC at public health facilities of Debre Birhan town, Ethiopia, 2023.**

satisfied; 5, very satisfied).1 point was given to satisfied and 0 points to unsatisfied. Those who were satisfied with ≥75% of the items were categorized as satisfied (those who responded as very satisfied, satisfied, or neutral), and those who were satisfied with <75% of the items were categorized as 'unsatisfied' (those who responded dissatisfied or very dissatisfied [29].

## Data quality assurance

Five diploma midwives participated as data collectors, and one BSC midwife controlled the overall activity of a data collection method as a supervisor. One day of training was given to data collectors on the objective of the study. Pre-testing on a similar set of respondents was done in Debre Sina Primary Hospital. It was done to check for reliability, appropriateness of format, and wording of the questionnaire.

## Data processing and analysis

Following completion of the data collection, questionnaires were checked for completeness and consistency, and data was entered using Epi Data version 4.6, then transferred to SPSS version 25 for analysis. Binary and multiple logistic regression analyses were performed. Variables with a p-value of 0.25 in the bi-variable analysis were considered for the multivariate analysis to control the effect of confounding variables. The odds ratio, along with a 95% confidence interval (CI), was computed to ascertain the strength of the association between the explanatory and outcome variables.

The regression model fitness was checked by the Hosmer-Lemshow goodness test (0.077) and Nagelkerke R square (0.463). The multi-collinearity assumption was checked by the variance inflation factor (VIF), and there is no multi-collinearity.

## Results

### Socio-Demographic characteristics of respondents

A total of 512 participants were enrolled in this study, with a response rate of 98%. The age of the mothers ranges from 18 to 45 years old, with a mean age of 32.9 years. The marital status of the participants revealed that 84.4% of them were married during the period of data collection. Moreover, 30.5% of respondents reported having completed primary education (Table 1).

### Obstetric characteristics of the respondents

Among the 512 respondents, 81.6% were multi-para. Of that, 63% were delivered at a health center during the previous childbirth. The majority (70%) of the respondents had planned pregnancy; 15.8% of participants had previous pregnancy-related complications; and 6.1% had current pregnancy-related complications. On the other hand, 46.8% of respondents reported having a close friend or a family member who has been delivered through a cesarean section. One-fourth of participants were not satisfied with previous intrapartum care (Table 2).

### Preference of caesarean delivery and their reasons for it

Women's preference for cesarean delivery was 26%, with a CI of 22.3%–29.9%. Of the mothers who preferred cesarean section as the mode of delivery, about one-fourth of participants were due to CS having less labor pain than vaginal delivery (Fig 2), (Table 3).

**Table 1. Socio demographic characteristics of pregnant women attending ANC in selected public health facilities in Debre Berhan, Ethiopia, 2023.**

| Variables | Category | Frequency | Percentage (%) |
|---|---|---|---|
| Residence | Urban | 337 | 65.8% |
| | Rural | 175 | 34.2% |
| Age | 18–25 | 113 | 22% |
| | 26–35 | 283 | 55.3% |
| | 36–45 | 116 | 22.7% |
| | Single | 37 | 7.2% |
| Marital status | Married | 431 | 84.2% |
| | Divorced | 31 | 6.1% |
| | Widowed | 13 | 2.5% |
| Income | >2500 | 145 | 28.3% |
| | 2500–4000 | 120 | 23.4% |
| | 4001–10000 | 135 | 26.4% |
| | >10000 | 112 | 21.9% |
| Occupation | Employed | 166 | 32.4% |
| | Un employed | 346 | 67.6% |
| Educational status | No formal education | 143 | 31.3% |
| | Primary education | 131 | 28.7% |
| | Secondary education | 67 | 14.7% |
| | College and above | 11 | 25.4% |
| Partners Educational status | No formal education | 143 | 22.7% |
| | Primary education | 131 | 13.1% |
| | Secondary education | 67 | 25.6% |
| | College and above | 116 | 13.1% |
| Occupational status | Employed | 174 | 34% |
| | Un employed | 338 | 66% |

## Factors associated with preference of caesareans delivery

The bivariate analysis showed that the knowledge of the respondents towards CS, previous satisfaction during intrapartum care, residence, marital status, occupation, planned pregnancy, previous spontaneous abortion, current pregnancy-related obstetric problem, and discussion with partners about the mode of delivery were factors associated with the preference of cesarean delivery (p-value less than 0.25) and added to the multivariable logistic regression analysis. In multivariate logistic regressions, previous satisfaction with intrapartum care, current pregnancy-related obstetric problems, knowledge of the respondents towards cesarean delivery, previous spontaneous abortion, and residence were significantly associated with preference for cesarean delivery (p-value less than 0.05).

The result showed that pregnant women who lived in urban residences were 1.9 times more likely to prefer CS as compared with women who lived in rural. (AOR = 1.9(1.03–3.5) P = 0.038*). Pregnant women who had previous abortions were 3 times more likely to prefer CS compared to pregnant women who had no previous spontaneous abortion(AOR = 3.1(1.5–6.3) P = 0.001*).

Pregnant women dissatisfied with previous intrapartum care preferred CS for the current pregnancy as a mode of delivery, the degree of preference increased 6 times as compared to women who were satisfied. (AOR = 6.3(3.58–11.29) P = 0.01*.

The other variable that was found to have a significant association was the knowledge of respondents about cesarean delivery, pregnant women who had no knowledge about cesarean delivery were 2.9 times more likely to prefer CS as compared to had knowledge about cesarean

**Table 2. Obstetric characteristics of pregnant women attending ANC in selected public health facilities in Debre Berhan, Ethiopia, 2023.**

| Variables | Category | Frequency | Percentage (%) |
|---|---|---|---|
| Previous history of spontaneous abortion | Yes | 78 | 15.2 |
| | No | 434 | 48.8 |
| Current number of ANC contact | First Visit | 234 | 45.7 |
| | Second Visit | 101 | 19.7 |
| | Third Visit | 49 | 9.6 |
| | 4th Visit | 46 | 9.0 |
| | > 4th visit | 82 | 16.0 |
| Parity | Primi para | 94 | 18.4 |
| | Multi | 418 | 81.6 |
| Previous pregnancy related problem | Yes | 66 | 12.3 |
| | No | 449 | 87.7 |
| Planed pregnancy | Yes | 352 | 68.8 |
| | No | 160 | 31.3 |
| Current pregnancy related problem | Yes | 36 | 7 |
| | No | 476 | 93 |
| Discussion with Partners | Yes | 306 | 59.8 |
| | No | 206 | 40.2 |
| Partner's support to preference | Yes | 294 | 57.4 |
| | No | 218 | 42.6 |
| Previous intra partum care satisfaction | Yes | 383 | 74.8 |
| | No | 129 | 25.2 |
| Attitude towards CD | Positive attitude | 369 | 70.9 |
| | Negative attitude | 149 | 29.1 |
| Knowledge towards CD | poor knowledge | 199 | 38.9 |
| | Intermediate | 109 | 21.3 |
| | knowledgeable | 204 | 39.8 |

delivery (AOR = 2.9(1.6–5.3)P = 0.01*). Pregnant mothers who have current pregnancy-related obstetric problems are 4.8 times more likely to prefer CS as compared to mothers who haven't.(AOR = 4.8, CI = (1.9–10),P = 0.001) (Table 4).

## Discussion

This study aimed to assess the preference for cesarean delivery and its associated factors among pregnant women attending antenatal care at public health facilities in Debrebrehan, Ethiopia. After analyzing the data, all associated variables were discussed with different literature, from local to global that showed the preference for cesarean sections.

Women's delivery preference is a subject that is widely researched and debated in many parts of the world. Women's autonomy, their satisfaction with childbirth, and their active participation in the decision-making process regarding the way they want to give birth to their children are becoming increasingly important. In Ethiopia, very little is known about women's preferences for delivery methods, and there is no evidence of mothers' preference for cesarean sections. Although CS rates in Ethiopia have also increased [30]. The purpose of this study was to assess the preference for cesarean delivery and its associated factors among pregnant women attending antenatal care at public health facilities to fill the aforementioned gap.

In this study, the prevalence of cesarean delivery among pregnant women attending ANC in public health facilities was 26%. Similar studies have been conducted in Ethiopia and other

# Preference of Caesarean Delivery

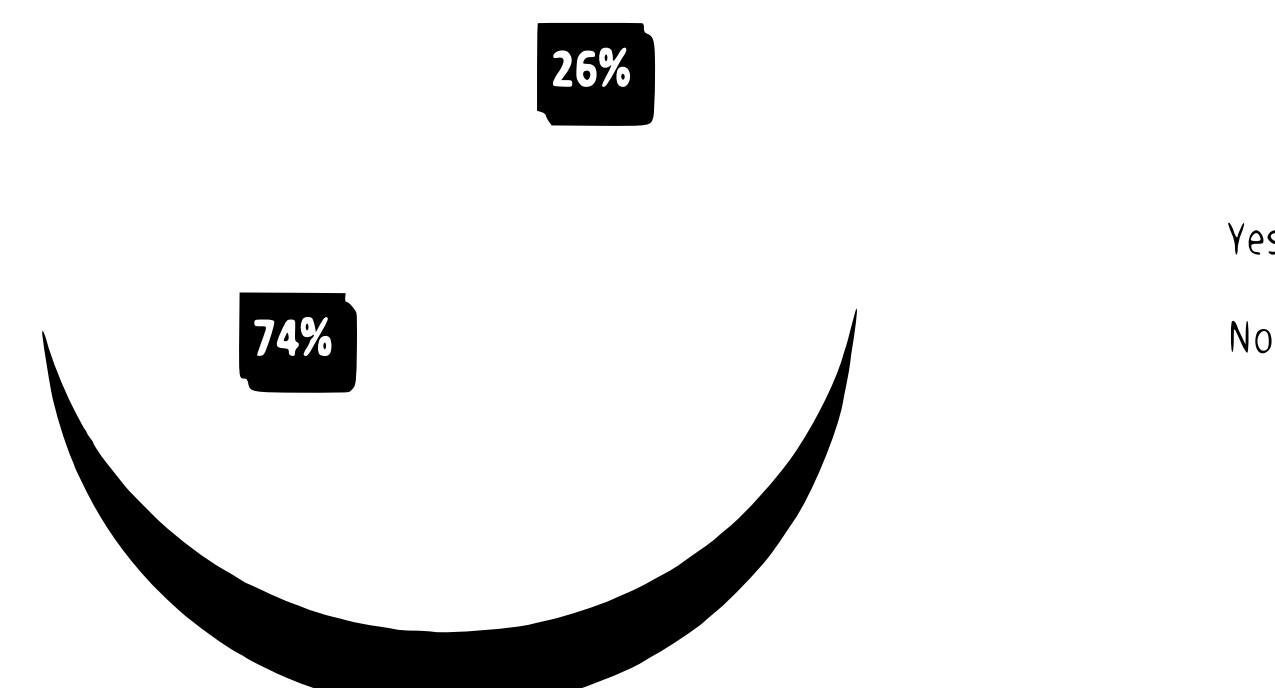

**Fig 2. Maternal preference of caesarean delivery among pregnant women attending ANC in selected public health facilities in Debre Berhan, Ethiopia, 2023.**

countries. This result is almost similar to the study in the southern part of Ethiopia and in Harer, which is 24.6% and 28%, respectively [21, 31]. The similarity may be that both studies were conducted during the ANC visit. However, this study result showed a higher preference for CD compared to other studies conducted at the University Hospital of Asyut, Egypt, in six European countries (Belgium, Iceland, Denmark, Estonia, Norway, and Sweden), where the preference for C/S was 12.2% [22, 32]. The discrepancy emerged when this study interviewed mothers in the ANC unit while the reference studies were conducted in delivery units, which may reduce the tendency of mothers to choose CD because of fear of childbirth.

In this study, dissatisfaction with intrapartum care was significantly associated with a preference for cesarean delivery. This was in line with studies conducted in Debre Markos, Gamo Gofa Zone, and southwest Ethiopia [33–35]. This could be because those who gave birth through spontaneous vaginal delivery may be experiencing labor pains. However, for those who have been delivered by CS, maybe the anesthesia relieves the pain of labor, and the surgery results in satisfaction.

**Table 3. Reasons behind women's preference' for caesareans delivery among pregnant women attending ANC in selected public health facilities in Debre Berhan, Ethiopia, 2023.**

| Variable | Frequency | Percentage |
|---|---|---|
| CS has less Labour pain | 99 | 24.8 |
| Avoidance of emergency cesarean section | 22 | 5.5 |
| Safer for women | 69 | 17.3 |
| Less risk of fetal distress | 41 | 10.3 |
| A chance to choose specific date | 9 | 2.3 |
| quick restoration for sexual activity | 12 | 3.0 |
| A fashion | 42 | 10.5 |
| Prior negative experience from vaginal delivery | 55 | 13.7 |
| Health care providers were not encouraging and reassuring during previous vaginal delivery | 10 | 2.5 |
| Fear or the need to avoid episiotomy | 31 | 7.8 |
| Other | 9 | 2.3 |

Further, according to the findings, mothers who had previous spontaneous abortions were found to have a statistically significant relationship with their preference for CD. Mothers who had previous spontaneous abortions were three times more likely to prefer CD than mothers who hadn't had spontaneous abortions. This result was slightly similar to another study that was conducted in Iran (OR = 1.7) [36].

There is also an association between preference for cesarean delivery and residence. The findings showed that living in an urban settlement was significantly associated with the preference for CD. Respondents living in urban areas had higher odds for the preference of CD compared to rural dwellers (odd ratio of 1.9). This is in line with other studies conducted in Ghana, Nepal, and Bangalore [37–39]. These could be urban women who are more likely to be educated and hear about the CD and are also financially able to afford the increased costs of a CD. In addition, living in an urban settlement also improves access to quality medical facilities that are well-equipped to perform CS.

Pregnant mothers who have a current pregnancy-related obstetric problem were four times more likely to prefer CS as compared to mothers who didn't. This finding is almost similar to a study conducted in Hawassa [24]. This might be due to a fear of intrapartum complications.

Pregnant mothers who had no knowledge of cesarean delivery were two times more likely to prefer CS. This is in line with a study done in urban Nigeria and Iran, where mothers who had no knowledge preferred CD [36, 40]. This may be because pregnant mothers, with a lack of knowledge, couldn't fully appreciate the health risks of maternal and fetal complications of CD.

## Conclusion

In this study, the preference for cesarean delivery was higher as compared to the World Health Organization recommendation. One-fourth of pregnant mothers involved in the study preferred C/S as their mode of delivery. In the previous who was not satisfied with their previous intrapartum care had no knowledge about cesarean delivery, had a previous history of spontaneous abortion, had an urban residence, and had a current pregnancy-related problem were significantly associated with the preference for cesarean delivery.

Table 4. Factors associated with preference of caesarean delivery caesareans delivery among pregnant women attending ANC in selected public health facilities in Debre Berhan, Ethiopia, 2023.

| Variables | Category | Preference of cesarean deliver | | COR (95%CI) | AOR(95% CI) | P value |
|---|---|---|---|---|---|---|
| | | Yes | No | | | |
| **Knowledge of the respondents** | Poor Knowledge | 66 | 133 | 2.2(1.4–3.5) | 2.9(1.6–5.3) | 0.001* |
| | Intermediate | 30 | 79 | 1.71(0.9–2.9) | 2.4(0.22–5.0) | 0.312 |
| | Knowledgeable | 37 | 167 | 1 | 1 | |
| **Previous intra partum Satisfaction** | Satisfied | 73 | 56 | 7 (4.5–10) | 6.3(3.5–11) | 0.001* |
| | Not Satisfied | 60 | 323 | 1 | 1 | |
| **Residence** | Urban | 105 | 232 | 2.3(1.4–3.7) | 1.9(1.0–3.5) | 0.038* |
| | Rural | 28 | 147 | 1 | 1 | |
| **Occupation** | Employed | 64 | 102 | 2.5(1.6–3.7) | 0.5(0.26–1.07) | 0.77 |
| | Non Employed | 69 | 277 | 1 | 1 | |
| **Planed pregnancy** | Yes | 110 | 242 | 2.7(1.6–4.4) | 1.07(0.4–2.3) | 0.858 |
| | No | 23 | 137 | 1 | 1 | |
| **Previous history of spontaneous abortion** | Yes | 48 | 30 | 6.56(3.9–10.9) | 3.1(1.5–6.36) | 0.001* |
| | No | 85 | 349 | 1 | 1 | |
| **Maternal Education** | No formal Education | 19 | 135 | 0.1(0.6–0.21) | 0.10(0.03–1.02) | 0.112 |
| | Primary Education | 21 | 135 | 0.13(0.69–0.23) | 0.1(0.46–1.2) | 0.431 |
| | Secondary Education | 36 | 63 | 0.4(0.2–0.8) | 0.4(0.2–1.3) | 0.325 |
| | College and above | 57 | 46 | 1 | 1 | |
| **Current Pregnancy related problem** | Yes | 25 | 11 | 7.7(3.6–16) | 4.8(1.9–10) | 0.001* |
| | No | 108 | 368 | 1 | 1 | |
| **Discussion with partner** | Yes | 88 | 218 | 1.4(1.9–2.1) | 0.8(0.4–1.4) | 0.545 |
| | NO | 45 | 161 | 1 | 1 | |

## Recommendation

➢ **Health care providers** should provide health education to increase the knowledge of pregnant mothers about cesarean delivery.

➢ Clinicians who are working in the delivery room should improve their service provision by using patient-centered care to increase patient satisfaction.

➢ Nurses, midwives, and other stakeholders in obstetric care should give health education and proper counseling during antenatal care to women during cesarean sections, as well as birth preparedness and complication readiness.

➢ **MOH**: designing strategies to enhance maternal satisfaction by strengthening adherence of health professional to intrapartum care.

➢ **Researchers:** A qualitative study is also required to better understand women's perspectives toward the preference for cesarean delivery, especially among mothers who had a pregnancy-related complication.

## Strength and Limitations of the study

➢ The main strength of this study is that try to incorporate variables that were not in previous studies. Like previous satisfaction on intra partum care, knowledge towards cesarean delivery. The study share the limitation of cross-sectional study.

## Supporting information

**S1 File. English version questioner.**
(DOCX)

**S1 Checklist. Clinical study checklist.**
(DOCX)

**S2 Checklist. STROBE checklist.**
(DOC)

**S1 Data. SPSS data.**
(SAV)

## Acknowledgments

The authors would like to thank and sincerely appreciate the study participants, data collectors, supervisors, and Debre-Berhan city health Administrative.

## Author Contributions

**Conceptualization:** Lemlem Zewudu.

**Data curation:** Lemlem Zewudu.

**Formal analysis:** Lemlem Zewudu.

**Funding acquisition:** Lemlem Zewudu, Fetene Keshaun, Mulualem Silesh, Mitiku Tefera, Eyob Ketema Bogale, Aberham Demis, Zewedie Yeshaw Tekle.

**Investigation:** Lemlem Zewudu, Fetene Keshaun, Mulualem Silesh, Mitiku Tefera, Eyob Ketema Bogale, Aberham Demis, Zewedie Yeshaw Tekle.

**Methodology:** Lemlem Zewudu, Fetene Keshaun, Mulualem Silesh, Mitiku Tefera, Eyob Ketema Bogale, Aberham Demis, Zewedie Yeshaw Tekle.

**Project administration:** Lemlem Zewudu, Fetene Keshaun, Mulualem Silesh, Mitiku Tefera, Eyob Ketema Bogale, Aberham Demis, Zewedie Yeshaw Tekle.

**Resources:** Lemlem Zewudu, Fetene Keshaun, Mulualem Silesh, Mitiku Tefera, Eyob Ketema Bogale, Aberham Demis, Zewedie Yeshaw Tekle.

**Software:** Lemlem Zewudu, Fetene Keshaun, Mulualem Silesh, Mitiku Tefera, Eyob Ketema Bogale, Aberham Demis, Zewedie Yeshaw Tekle.

**Supervision:** Lemlem Zewudu, Fetene Keshaun, Mulualem Silesh, Mitiku Tefera, Eyob Ketema Bogale, Aberham Demis, Zewedie Yeshaw Tekle.

**Validation:** Lemlem Zewudu, Fetene Keshaun, Mulualem Silesh, Mitiku Tefera, Eyob Ketema Bogale, Aberham Demis, Zewedie Yeshaw Tekle.

**Visualization:** Lemlem Zewudu, Fetene Keshaun, Mulualem Silesh, Mitiku Tefera, Eyob Ketema Bogale, Aberham Demis, Zewedie Yeshaw Tekle.

**Writing – original draft:** Lemlem Zewudu, Fetene Keshaun, Mulualem Silesh, Mitiku Tefera, Eyob Ketema Bogale, Aberham Demis, Zewedie Yeshaw Tekle.

**Writing – review & editing:** Lemlem Zewudu, Fetene Keshaun, Mulualem Silesh, Mitiku Tefera, Eyob Ketema Bogale, Aberham Demis, Zewedie Yeshaw Tekle.

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
