## [Decision Letter · Decision Letter 0]

12 Oct 2023

PONE-D-23-28572PREFERENCE OF CESAEREAN DELIVERY AND ITS ASSOCIATED FACTORS AMONG PREGNANT WOMEN ATTENDING ANTE NATAL CARE AT PUBLIC HEALTH FACILITIES OF DEBREBRHAN CITY, ETHIOPIA, 2023.PLOS ONE

Dear Dr. Tefera,

Thank you for submitting your manuscript to PLOS ONE. After careful consideration, we feel that it has merit but does not fully meet PLOS ONE’s publication criteria as it currently stands. Therefore, we invite you to submit a revised version of the manuscript that addresses the points raised during the review process.

We look forward to receiving your revised manuscript.

Kind regards,

Abera Mersha, MSc.

Academic Editor

PLOS ONE

- https://doi.org/10.1155/2021/1751578

- https://doi.org/10.2147/RMHP.S330932

- https://doi.org/10.5539/gjhs.v13n11p89

- ttps://dx.doi.org/10.4314/ahs.v22i2.56

In your revision ensure you cite all your sources (including your own works), and quote or rephrase any duplicated text outside the methods section. Further consideration is dependent on these concerns being addressed.

Reviewers' comments:

Reviewer's Responses to Questions

**Comments to the Author**

1. Is the manuscript technically sound, and do the data support the conclusions?

Reviewer #1: Partly

Reviewer #2: Yes

Reviewer #3: Yes

2. Has the statistical analysis been performed appropriately and rigorously? 

Reviewer #1: Yes

Reviewer #2: Yes

Reviewer #3: Yes

3. Have the authors made all data underlying the findings in their manuscript fully available?

Reviewer #1: Yes

Reviewer #2: No

Reviewer #3: Yes

4. Is the manuscript presented in an intelligible fashion and written in standard English?

Reviewer #1: No

Reviewer #2: No

Reviewer #3: Yes

5. Review Comments to the Author

Reviewer #1: 1. Stages of pregnancy might be considered because the intention for preferring mode of delivery may be different in the 1st, 2nd, and 3rd trimester of pregnancy. If a first-trimester pregnant mother were included in this study, the criteria for diagnosing a nonviable pregnancy should be included.

2. The tables in this manuscript are not standardized tables. The tables in the manuscript need formatting for readability.

3. The manuscript needs language editing, especially grammatical errors, which should be corrected by consulting a language expert or using a grammar checker.

4. There is no coherence between the result, conclusion, and recommendations. Unrelated recommendations are written in this manuscript.

Reviewer #2: 1. Exclusion criteria: whay were the mother who had cesearean section prior excluded? please explain why they were

excluded

2. please make the table of bivariate and multivariate in the results section, then explain how the significant factors would explain the findings

3. get help with editing the essay part of the English before sending it for publication, for better understanding

4. Provide a point on the recomendation that adresses the concerns of clinicians

5. provide recomendatoions guided by the WHO guidelines and the Ministry of Health

Reviewer #3: Work on the background comments on the manuscript

Work on the ethical considerations as presented including using names of health facilities assessed

Also look at the methodology to tweak a few areas

Work on the results and discussions attached in the manuscript reviewed attached

6. PLOS authors have the option to publish the peer review history of their article (what does this mean?). If published, this will include your full peer review and any attached files.

Reviewer #1: No

Reviewer #2: No

Reviewer #3: No

---

## [Author Response · Author response to Decision Letter 0]

10 Nov 2023

We have revised & ready to the next work

---

## [Decision Letter · Decision Letter 1]

13 Dec 2023

PONE-D-23-28572R1Preference of Cesarean Delivery and Its Associated Factors Among Pregnant Women Attending Ante Natal Care at Public Health Facilities of Debrebrehan City, Ethiopia: Cross-Sectional StudyPLOS ONE

Dear Dr. Tefera,

Thank you for submitting your manuscript to PLOS ONE. After careful consideration, we feel that it has merit but does not fully meet PLOS ONE’s publication criteria as it currently stands. Therefore, we invite you to submit a revised version of the manuscript that addresses the points raised during the review process.

We look forward to receiving your revised manuscript.

Kind regards,

Abera Mersha, MSc.

Academic Editor

PLOS ONE

Reviewers' comments:

Reviewer's Responses to Questions

**Comments to the Author**

1. If the authors have adequately addressed your comments raised in a previous round of review and you feel that this manuscript is now acceptable for publication, you may indicate that here to bypass the “Comments to the Author” section, enter your conflict of interest statement in the “Confidential to Editor” section, and submit your "Accept" recommendation.

Reviewer #3: All comments have been addressed

Reviewer #4: (No Response)

2. Is the manuscript technically sound, and do the data support the conclusions?

Reviewer #3: Yes

Reviewer #4: Partly

3. Has the statistical analysis been performed appropriately and rigorously? 

Reviewer #3: Yes

Reviewer #4: Yes

4. Have the authors made all data underlying the findings in their manuscript fully available?

Reviewer #3: Yes

Reviewer #4: Yes

5. Is the manuscript presented in an intelligible fashion and written in standard English?

Reviewer #3: Yes

Reviewer #4: No

6. Review Comments to the Author

Reviewer #3: The author addressed the comments as expected and improved the manuscript significantly. However, I would request the author address the few comments and the paper can be accepted, mostly grammatical and formatting issues

Reviewer #4: The authors conducted cross sectional study to determine preference of caesarean delivery among pregnant women in Debre Berhan city. They have recruited 512 participants with multistage sampling. My comments here

1. single population proportion formula was used to calculate the sample size; however, the study was exploring factors associated with preference for caesarean delivery. Here the sample size for the factors was not estimated yet multivariate logistic regression was fitted. Hence adequately representative sample may not be used.

2. The exclusion criteria:

how was the outcome of previous pregnancy considered?

what was the Gestational age limit used to delineate between 1st, 2nd and 3rd trimester.

How was the Gestational age assessment?

how were women with prior home delivery entertained?

3. In the knowledge assessment there is inconsistent use of terms like poor knowledge, no knowledge,

4. The information, counselling given Durning ANC varies by trimester and gestational age i.e. mode of delivery often discussed in the third trimester, that could potentially affect the knowledge on caesarean section. How do you see treating (including in the study) participants in the first and third trimester, those with first ANC contact with those having more than one ANC Contact.

5. The authors didn't mention the limitation of the study, could you elaborate.

6. There are flaws in the Language used, I recommend looking into it again

7. PLOS authors have the option to publish the peer review history of their article (what does this mean?). If published, this will include your full peer review and any attached files.

Reviewer #3: **Yes: **Emmanuel Ekung

Reviewer #4: No

---

## [Author Response · Author response to Decision Letter 1]

19 Dec 2023

We are try to revise based on the given comments and ready to do the next comments.

---

## [Decision Letter · Decision Letter 2]

27 Dec 2023

Preference of Cesarean Delivery and Its Associated Factors Among Pregnant Women Attending Ante Natal Care at Public Health Facilities of Debrebrehan City, Ethiopia: Cross-Sectional Study

PONE-D-23-28572R2

Dear Dr. Tefera,

We’re pleased to inform you that your manuscript has been judged scientifically suitable for publication and will be formally accepted for publication once it meets all outstanding technical requirements.

Kind regards,

Abera Mersha, MSc.

Academic Editor

PLOS ONE

Additional Editor Comments (optional):

Reviewers' comments:

Reviewer's Responses to Questions

**Comments to the Author**

1. If the authors have adequately addressed your comments raised in a previous round of review and you feel that this manuscript is now acceptable for publication, you may indicate that here to bypass the “Comments to the Author” section, enter your conflict of interest statement in the “Confidential to Editor” section, and submit your "Accept" recommendation.

Reviewer #4: All comments have been addressed

2. Is the manuscript technically sound, and do the data support the conclusions?

Reviewer #4: Yes

3. Has the statistical analysis been performed appropriately and rigorously? 

Reviewer #4: Yes

4. Have the authors made all data underlying the findings in their manuscript fully available?

Reviewer #4: Yes

5. Is the manuscript presented in an intelligible fashion and written in standard English?

Reviewer #4: Yes

6. Review Comments to the Author

Reviewer #4: The authors have significantly improved the manuscript, they tried to justify the sample size calculation. They tried to explain the inclusion and exclusion criteria. However, the limitation of the study is not explicitly elaborated.

7. PLOS authors have the option to publish the peer review history of their article (what does this mean?). If published, this will include your full peer review and any attached files.

Reviewer #4: No

---

## [Editor Report · Acceptance letter]

21 Jan 2024

PONE-D-23-28572R2 

PLOS ONE

Dear Dr. Tefera, 

I'm pleased to inform you that your manuscript has been deemed suitable for publication in PLOS ONE. Congratulations! Your manuscript is now being handed over to our production team.

Kind regards, 

on behalf of

Mr. Abera Mersha 

Academic Editor

PLOS ONE